

# Gas Reference Materials for Underpinning Atmospheric Measurements of Stable Isotopes of Nitrous Oxide

Ruth E. Hill-Pearce[1], Aimee Hillier[1], Eric Mussell Webber[1], Kanokrat Charoenpornpukdee[1,2], Simon O'Doherty[2], Joachim Mohn[3], Christoph Zellweger[3], David R. Worton[1] and Paul J. Brewer[1]

[1] National Physical Laboratory, Hampton Road, Teddington, TW11 0LW, United Kingdom

[2] University of Bristol, Atmospheric Chemistry Research Group, Bristol, United Kingdom

[3] Empa, Swiss Federal Laboratories for Materials Science and Technology, Laboratory for Air Pollution /Environmental Technology, Dübendorf, Switzerland

*Correspondence to*: ruth.pearce@npl.co.uk

**Abstract.** The precise measurement of the amount fraction of atmospheric nitrous oxide ($N_2O$) is required to understand global emission trends. Analysis of the site-specific stable isotopic composition of $N_2O$ provides a means to differentiate emission sources. The availability of accurate reference materials of known $N_2O$ amount fractions and isotopic composition is critical for achieving these goals. We present the development of nitrous oxide gas reference materials for underpinning measurements of atmospheric composition and isotope ratio. Uncertainties target the World Metrological Organisation Global Atmosphere Watch (WMO-GAW) compatibility goal of 0.1 nmol mol[-1] and extended compatibility goal of 0.3 nmol mol[-1], for atmospheric $N_2O$ measurements in an amount fraction range of 325-335 nmol mol[-1]. We also demonstrate the stability of amount fraction and isotope ratio of these reference materials and present a characterisation study of the cavity ring down spectrometer used for analysis of the reference materials.

## 1 Introduction

Nitrous oxide ($N_2O$) is a greenhouse gas with a global warming potential approximately 265 times that of carbon dioxide ($CO_2$) (Myhre et al., 2013). Atmospheric amount fractions of $N_2O$ are increasing at a rate of ~0.36 % per year (WMO, 2019b). Recent measurements of $N_2O$ in the unpolluted troposphere are in an amount fraction range of 325-335 nmol mol[-1] (WMO, 2019a).





Current amount fractions of 332 nmol mol$^{-1}$ have been published by the WMO (WMO, 2020). The growing field of N$_2$O research is focused on improving understanding of the global N$_2$O budget. A comprehensive identification of the N$_2$O sources and sinks, and the contribution of each to the global N$_2$O budget is required for N$_2$O mitigation studies (Lewicka-Szczebak et al., 2014).


Anthropogenic N$_2$O is released into the atmosphere mainly via multiple reaction pathways from soil and marine sources as a result of fertiliser use, aside technical emissions from industrial and combustion processes (Snider et al., 2015; Kantnerová et al., 2019; Toyoda et al., 2017). These different sources emit N$_2$O with distinct isotopic compositions, which can be used as an isotopic signature or fingerprint for identification (Denk et al., 2017). The most abundant N$_2$O isotopocules are: $^{14}$N$^{14}$N$^{16}$O, $^{14}$N$^{14}$N$^{18}$O, $^{14}$N$^{15}$N$^{16}$O and $^{15}$N$^{14}$N$^{16}$O. The site specific isotopomers display $^{15}$N substitution at the central, alpha ($\alpha$, $^{14}$N$^{15}$N$^{16}$O) and the terminal, beta ($\beta$, $^{15}$N$^{14}$N$^{16}$O) position.


Isotopic abundances are given in the delta notation ($\delta$) and expressed as the amount fraction ratio ($x$) of minor to major isotopic species in a sample (R$_{sample}$), relative to a reference value (R$_{reference}$). For $\delta^{15}$N, the isotope abundance scale is AIR-N$_2$, for $\delta^{18}$O, VSMOW (Vienna Standard Mean Ocean Water) (Toyoda and Yoshida, 1999). As differences in isotopic composition between sample and reference are usually small, delta values are generally expressed in per mille (‰).


$$R = \frac{x(^{15}N)}{x(^{14}N)} \tag{1}$$


$$\delta^{15}N[‰] = \left( \frac{R_{sample}}{R_{reference}} - 1 \right) \tag{2}$$

The bulk $\delta^{15}$N is representative of the average $\delta^{15}$N value, as expressed in equation 3.


$$\delta^{15}N^{bulk} = \left( \delta^{15}N^{\alpha} + \delta^{15}N^{\beta} \right)/2 \tag{3}$$

The site-specific distribution of $^{15}$N can be expressed by the site preference (SP), defined as the difference in $^{15}$N between $\alpha$ and $\beta$ position, displayed in equation 4 (Yoshida and Toyoda, 2000).


$$SP = \delta^{15}N^{\alpha} - \delta^{15}N^{\beta} \tag{4}$$





Isotopologue quantification is well established for atmospheric $CO_2$ (Flores et al., 2017). However, quantification of $N_2O$ isotopocules proves a greater analytical challenge due to: (i) substantially lower atmospheric amount fractions, (ii) analytical difficulties in the provision of position specificity of the standard technique, isotope ratio mass spectrometry (IRMS), due to difficulty in the application of correction factors to account for the re-arrangement of $^{15}N$ and $^{14}N$ within the ion source (Mohn et al., 2014) and (iii) the lack of internationally accepted $N_2O$ isotope reference materials with stated uncertainty (Ostrom et al., 2018).


In summary, this results in a limited compatibility of laboratories analyses for $N_2O$ isotope measurements (Mohn et al., 2014). In turn, an improvement in standardisation of the assignment of delta values, within and between laboratories, can only be achieved through calibration with accurate isotope ratio reference materials.

Atmospheric $N_2O$ has a relative abundance of 0.9903 mol mol$^{-1}$ for the major isotopolocule $^{14}N^{14}N^{16}O$. The minor isotopocules $^{14}N^{15}N^{16}O$, $^{15}N^{14}N^{16}O$ and $^{14}N^{14}N^{18}O$ display a relative abundance of 3.64 x 10$^{-3}$, 3.64 x 10$^{-3}$ mol mol$^{-1}$ and 1.99 x 10$^{-3}$ mol mol$^{-1}$ respectively (Kantnerová et al., 2019). This corresponds to less than 1 nmol mol$^{-1}$ for $^{14}N^{14}N^{18}O$ in ambient amount fraction $N_2O$ reference materials. High sensitivity instrumentation is required to precisely quantify the low amount fractions of the minor $N_2O$ isotopocules (Griffith et al., 2012).


Recent advances in spectroscopic instrumentation have improved $N_2O$ isotopolocule quantification. Cavity Ring-Down Spectroscopy (CRDS) has been applied to the real-time amount fraction and isotopic composition measurements of $N_2O$ in ambient air. In the laboratory, this technique has demonstrated a precision of < 0.05 nmol mol$^{-1}$ for $N_2O$ amount fraction and < 0.5 ‰ for $\delta^{15}N^\alpha$, $\delta^{15}N^\beta$ and $\delta^{18}O$-$N_2O$ with 5-minute averaging times (Harris et al., 2020; Erler et al., 2015). This precision

is comparable with Off-Axis Integrated Cavity Output Spectroscopy analysis (OA-ICOS) (Van Der Schoot et al., 2015), which demonstrated in-field analytical precisions of < 0.07 nmol mol$^{-1}$, and superior to the standard technique for $N_2O$ amount fraction, gas chromatography with electron capture detector (GC-ECD) (Lopez et al., 2015). For $\delta^{15}N^\alpha$ and $\delta^{15}N^\beta$ the performance of CRDS is approaching IRMS (Ostrom et al., 2018).

Advances in instrumentation must be coupled with advances in high precision isotope ratio reference materials, particularly for the calibration of the site specific isotopic composition $\delta^{15}N^\alpha$ and $\delta^{15}N^\beta$, to achieve accurate calibration of the small variations in isotopocule abundances observable in ambient $N_2O$ (Ostrom and Ostrom, 2017). Isotope ratio reference materials are required which span the full range expected in ambient $N_2O$ samples and covered by the World Metrological Organisation (WMO) scale (260-370 nmol mol$^{-1}$). The currently available pure $N_2O$ secondary reference materials USG5S1 and USG5S2

(Reston Stable Isotope Laboratory) differ in their $\delta^{15}N^\alpha$ and $\delta^{15}N^\beta$ values but span a narrow range of $\delta^{15}N$ and $\delta^{18}O$ values (< 1 ‰) limiting applicability for use as calibration materials (Ostrom et al., 2018). The availability of $N_2O$ isotope ratio reference materials has the potential to improve calibration of analytical instrumentation and increase interlaboratory agreement.

Crucial for the development of reference materials is the stability of the $N_2O$ composition and isotope ratio. (Ganesan et al.,
2013) reported no significant drift in amount fraction for a nominally 325 nmol mol$^{-1}$ $N_2O$ in compressed air reference material
in an aluminium cylinder (Scott Marrin) over a three-year period. Similar findings were reported by (Lushozi et al., 2019) but
no study is available yet on the stability of the $N_2O$ isotope ratio at ambient amount fractions. In addition, improvements in the
preparation and availability of $N_2O$ reference materials at ambient amount fraction is required to achieve the challenging
WMO-GAW compatibility goals (Brewer et al., 2019).


We present work on the development of $N_2O$ reference materials for underpinning atmospheric composition and isotope ratio
with uncertainties targeting the WMO-GAW compatibility goals. We describe the characterisation of precision, repeatability
and drift of a CRDS laser spectrometer. We also present work on all elements of the preparation process such as gravimetry,
purity analysis, validation, stability, passivation of storage media, and matrix effects. These developments are extended to
multi-components mixtures of $N_2O$ with other greenhouse gases ($CO_2$, $CH_4$ and CO) in a synthetic air matrix containing
atmospheric amount fractions of argon, oxygen and nitrogen, as required for calibration of spectroscopic instruments for
atmospheric measurements.

## 2 Experimental

### 2.1 Gravimetric preparation of primary reference materials (PRMs)

All Primary Reference Materials (PRMs) were prepared by gravimetry, in accordance with ISO 6142-1:2015, in 10 L
aluminium cylinders (Luxfer) with a range of outlet diaphragm valves (Ceodeux): BS341 no. 14, DIN 477 no. 1 and DIN 447
no. 8. The cylinders were treated internally by electropolishing (Luxfer) or with a range of proprietary passivation processes
including SpectraSeal™ (BOC), Megalong™ and Aculife IV/ III™ (Air Liquide) to inhibit adsorption of target components.
Cylinders were evacuated using an oil-free pump (Scrollvac SC15D, Leybold Vacuum) and turbo molecular pump with
magnetic bearing (Turbovac 340M, Leybold Vacuum) to a pressure of $< 3 \times 10^{-7}$ mbar. Synthetic air was gravimetrically
produced by blending argon (BIP+, Air Products), oxygen (N6.0, BOC), and nitrogen (BIP+, Air Products) to match
atmospheric amount fractions (0.94, 20.96, and 78.10 cmol mol$^{-1}$, respectively) (Tohjima et al., 2009). The purity of the matrix
gas was assessed for amount fraction of $N_2O$ as detailed below.

The reference materials were produced gravimetrically by the addition of $N_2O$ (5.0, Air Liquide) via a transfer vessel (capped
¼" diameter tube, with a nominal volume of 45 mL, Swagelok, electro-polished stainless steel). The transfer vessel was



weighed against a tare vessel matched for size and shape before and after $N_2O$ addition into the evacuated cylinder (Mettler-
Toledo XP2004S, $\pm$ 0.3 mg). Filling via a transfer vessel was used to achieve low uncertainty on the addition of small masses.
Nitrogen was added via direct addition to the cylinder, through purged $^1/_{16}$" tubing (Swagelok, electro-polished stainless steel)
to produce reference materials with nominal $N_2O$ amount fractions of 500 $\mu$mol mol$^{-1}$. The mass of nitrogen added was
determined by weighing of the cylinder before and after addition against a tare cylinder on an automatic weighing facility,
developed by the Korean Research Institute for Standards and Science (KRISS) (Mettler-Toledo XP26003L, $\pm$ 3 mg) (Lim et
al., 2017).

Atmospheric amount fraction reference materials in the range 300-360 nmol mol$^{-1}$ $N_2O$ were prepared by the addition of
nominally 500 $\mu$mol mol$^{-1}$ $N_2O$ reference materials via a transfer vessel into an evacuated 10 L cylinder and dilution with
synthetic air by the direct addition of argon, oxygen and nitrogen at atmospheric amount fractions (Tohjima et al., 2009). The
cylinder was weighed before and after each matrix gas addition. Argon was added to the cylinder from a nominally 30 % argon
in nitrogen pre-mixture cylinder.

Gravimetric uncertainties associated with the preparation of $N_2O$ reference materials were calculated according to the *Guide
to the Expression of Uncertainty in Measurement* (BIPM et al., 2008). The gravimetric uncertainty associated with the $N_2O$
amount fraction in the prepared mixtures is determined by the software package gravcalc2 (Brown, 2009), which combines
the uncertainty in relative molar mass, the uncertainty in the mass of the parent mixture addition, and the uncertainty in the
amount fraction of $N_2O$ in the parent mixture according to ISO 6142 (ISO, 2015).

The total gravimetric uncertainty of the reference material combines gravimetric uncertainty from the weighing data with
uncertainty in the amount of $N_2O$ in the matrix gases. For low amount fraction reference materials, the total uncertainty can be
dominated by uncertainty in accurately quantifying trace amount fractions of the compound of interest within the matrix gases.
As such a careful analysis of the trace $N_2O$ in the matrix gases is required. The amount fraction of $N_2O$ in the matrix was
determined by standard addition of a 325 nmol mol$^{-1}$ reference material into a synthetic air prepared with the same argon in
nitrogen premix and the same oxygen and nitrogen cylinders. This method is as described in (Hill-Pearce et al., 2018). The
zero offset of the analyser was determined by removal of trace $N_2O$ in the matrix gas (SAES Getter/Entegris PS15 GC-50).

## 2.2 Preparation of reference materials for studying the influence of pressure on composition

Aluminium cylinders (0.85 L, Luxfer) were filled to 30-35 bar with a nominally 325 nmol mol$^{-1}$ $N_2O$ in synthetic air reference
material. Within one week of filling, the cylinders were sampled at an excess flow rate of 0.5 L min$^{-1}$ into the CRDS analyser

(Picarro G5131-*i*), until the cylinders were at nominally ambient pressure. The cylinder pressure was monitored during sampling with a pressure transducer (Omega PXM 319) and data was recorded via LabVIEW.

In a second approach, nominally 325 nmol mol$^{-1}$ N$_2$O in synthetic air reference materials were prepared from the same 500 µmol mol$^{-1}$ N$_2$O reference material in 10 L cylinders with three different commercially available internal passivation processes.

The cylinders were sampled into the CRDS analyser following the same procedure as for the 0.85 L cylinders.

## 2.3 Analytical methods

A cavity ring-down spectrometer (Picarro G5131-*i*) was used for the analysis of the ambient amount fraction N$_2$O mixtures. The instrument allows simultaneous monitoring of N$_2$O amount fraction and isotopic composition through measurement of the

bulk $\delta^{15}$N, $\delta^{18}$O and the site-specific $\delta^{15}$N$^\alpha$ and $\delta^{15}$N$^\beta$. Bulk $\delta^{15}$N is calculated as the average of the site-specific $\delta^{15}$N$^\alpha$ and $\delta^{15}$N$^\beta$. The instrument comprises an internal pump and a critical orifice to reduce the gas flow into the cavity of the analyser. An excess flow was provided to the instrument (0.5 L min$^{-1}$) and the excess vented to the atmosphere to ensure stable (atmospheric) inlet pressure and no contamination with ambient air.

Analysis of the amount fraction of argon in the nominally 30 % argon in nitrogen pre-mixture cylinders was performed by gas chromatography with thermal conductivity detector (GC-TCD; Agilent 6890) using a capillary column (Molsieve 5A, 30 m x 0.53 mm x 0.50 µm) operated isothermally at 30 ± 1 °C.

## 3 Characterisation of the CRDS

## 3.1 Analytical uncertainty

### 3.1.1 Allan deviation

A 325 nmol mol$^{-1}$ N$_2$O in synthetic air reference material was analysed continuously over 25 hours, collecting temporal trends of N$_2$O amount fractions and isotope delta values. The Allan deviation was calculated, to assess the optimum averaging time and the maximum precision (Figure 1).



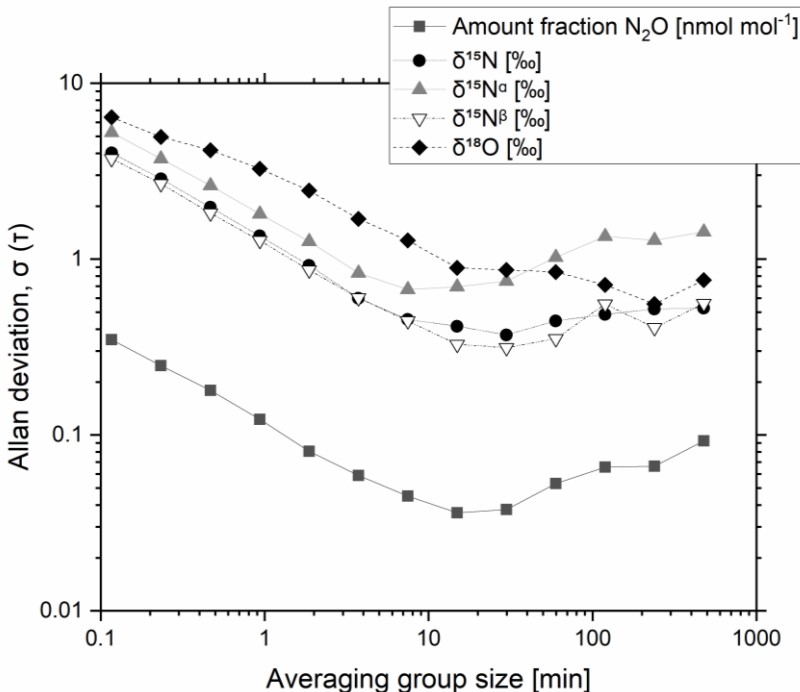

**Figure 1: Typical Allan deviation plot for a CRDS N₂O isotope analyser (G5131-*i*) as a function of averaging time for a nominally 325 nmol mol⁻¹ N₂O reference material.**


The Allan deviation initially decreases with an increase in the averaging time and reaches a minimum for $N_2O$ amount fractions (0.036 nmol mol⁻¹) and delta values $\delta^{15}N$ (0.37 ‰), $\delta^{15}N^\alpha$ (0.67 ‰), $\delta^{15}N^\beta$ (0.33 ‰) and $\delta^{18}O$ (0.89 ‰) for averaging times of around 15 minutes. For longer averaging times, an increase in the Allan deviation is shown and likely to be a result of analyser drift. An averaging time of 10 minutes was adopted to ensure both minimal uncertainty for comparing the reference gas to a sample gas, and efficient use of the reference material. Achieved precisions for $N_2O$ amount fraction and isotope ratios are in agreement with the typical precisions reported by Picarro in the instrument specification of < 0.05 nmol mol⁻¹ $N_2O$ and < 0.7 ‰ for $\delta^{15}N$, $\delta^{15}N^\alpha$, $\delta^{15}N^\beta$, $\delta^{18}O$ for a 10-minute averaging period (Picarro, 2017).



**3.2 Characterisation of the CRDS for reported delta values with N₂O amount fraction.**

The characterisation of the CRDS for reported delta values with $N_2O$ amount fraction was assessed with both statically and dynamically generated reference materials. Dynamic reference materials were produced in the amount fraction range 150-1100 nmol mol⁻¹ by dilution from a nominally 320 µmol mol⁻¹ $N_2O$ in synthetic air reference material with synthetic air using a dynamic dilution device comprising one diluent and three standard critical flow orifices (Hill-Pearce et al., 2018). The static






and dynamic reference materials were generated alternately for 4 iterations, with synthetic air measured between each set. Due to the large number of measurements recorded, a reduced sampling time of 5 minutes was adopted for each measurement interval resulting in a slightly lower standard deviation of 0.03 nmol mol$^{-1}$ for amount fractions.


### 3.3 Delta $^{15}$N

The $\delta^{15}$N values analysed by the G5131-$i$ analyser were recorded for each static and dynamic reference material for four repetitions of five minutes. The mean value of the stable response was calculated. The change in reported delta value with

amount fraction was assessed and found to vary with a linear function with respect to the reciprocal of N$_2$O amount fraction as reported by (Harris et al., 2020) for the same CRDS model, with a different year of manufacture. Figure 2 shows the CRDS analyser response to $\delta^{15}$N for static and dynamic reference materials prepared from the same pure N$_2$O source in the amount fraction range of 300-1500 nmol mol$^{-1}$. (Winther et al., 2018) reported the same trend for dependence of reported $\delta^{15}$N on N$_2$O amount fraction, attributing the amount fraction dependence to offsets in the measurement of $^{14}$N$^{15}$N$^{16}$O and $^{15}$N$^{14}$N$^{16}$O. The

agreement between static and dynamic reference materials is discussed in the results section.

### 3.4 Delta $^{18}$O

Similarly, the $\delta^{18}$O channel response was recorded for static and dynamic reference materials. No variation in reported $\delta^{18}$O was observed between the analyser response of the static and dynamic reference materials beyond the measurement uncertainty

for N$_2$O amount fractions over the range of 300-1500 nmol mol$^{-1}$ (Figure 2). (Harris et al., 2020) reported a linear increase in $\delta^{18}$O of ~4 ‰ with the reciprocal of N$_2$O amount fraction over a similar N$_2$O amount fraction range for the same model of analyser (2015 model) but stated that the change in delta value with amount fraction might vary between different analysers of the same model.





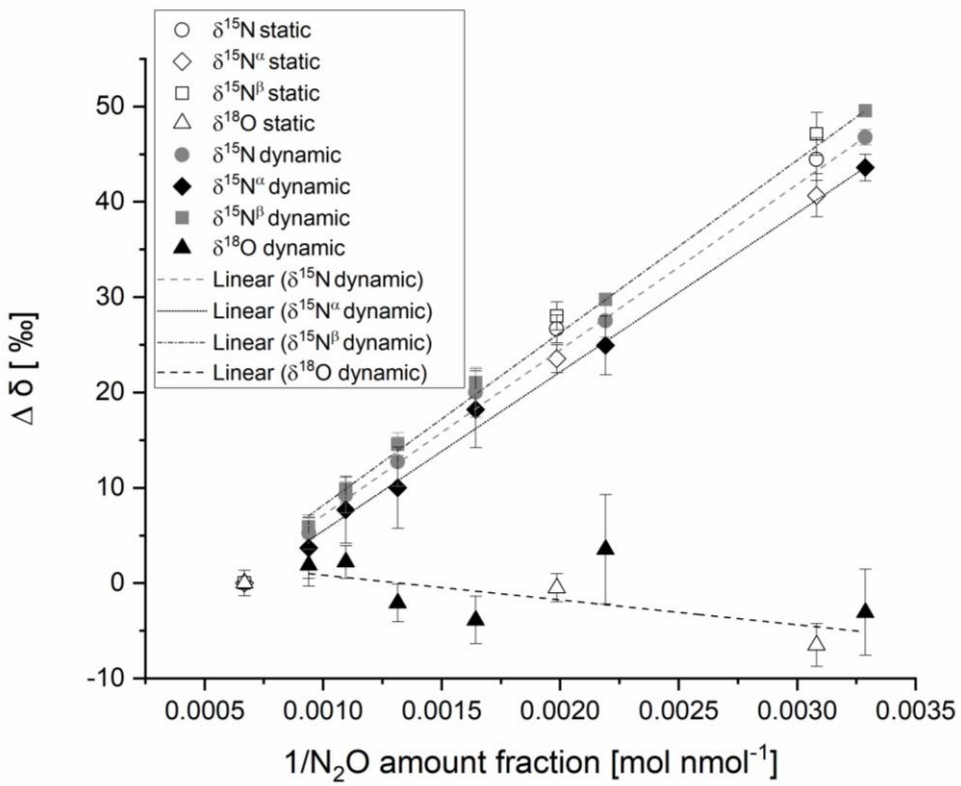

**Figure 2: CRDS analyser response for $\delta^{15}N$ (circles), $\delta^{15}N^{\alpha}$ (diamonds), $\delta^{15}N^{\beta}$ (squares) and $\delta^{18}O$ (triangles) with reciprocal of $N_2O$ amount fraction for static (open data labels) and dynamic (filled data labels) reference materials in the amount fraction range of 300-1500 nmol mol$^{-1}$. Error bars represent the repeatability in ‰ between the four repetitions of a five-minute average. The dotted and dashed lines represent the linear regression of dynamic reference materials for each isotopocule.**



The agreement in $\delta^{15}N$ and $\delta^{18}O$ between static and dynamic reference materials (shown in Figure 2) indicates minimal fractionation of isotopocules on dilution through a critical flow orifice based dynamic system or on production of the reference materials by filling though an intermediate vessel and dilution. No variation in reported delta values beyond the measurement uncertainty for $N_2O$ amount fractions over the range of 300-1500 nmol mol$^{-1}$ was observed between the analyser response of

the static and dynamic reference materials (Figure 2). However, the large uncertainty makes comparisons of the delta value between similar amount fractions challenging. The uncertainty would be reduced by increasing the averaging time.



## 4 Results and Discussion

### 4.1 Uncertainty in $N_2O$ amount fraction

Uncertainty in the amount fraction of $N_2O$ in a reference material has several sources including: uncertainty due to gravimetric
preparation (weighing uncertainties), uncertainty in the purity of the gases used (e.g. amount fraction of $N_2O$ in the matrix), cylinder effects such as adsorption of the gas molecules onto the walls of the cylinder and valve, uncertainties in amount fraction due to the stability of the gas reference material and analytical precision of the measurement technique. Each uncertainty contribution is discussed below.

**4.1.1 Uncertainty and reproducibility in the amount fraction of reference materials due to gravimetric production.**

To assess the uncertainty in the amount fraction from production of $N_2O$ reference materials, 8 reference materials were produced by two separate operators from two separate 500 µmol mol$^{-1}$ $N_2O$ in nitrogen reference materials but the same matrix gases and pure $N_2O$ source. Four of the reference materials were produced at nominally 337 nmol mol$^{-1}$ and four were produced at nominally 326 nmol mol$^{-1}$.


The combined contribution to the uncertainty due to gravimetry and purity of the components for the ambient amount fraction $N_2O$ in synthetic air reference materials produced, as detailed above, is 0.08 % ($k$=2) 0.28 nmol mol$^{-1}$. This uncertainty is within the WMO-GAW extended compatibility goals of ± 0.3 nmol mol$^{-1}$.

The combined expanded uncertainty is dominated by the uncertainty in the mass of parent gas additions. There is a 0.3 mg uncertainty on the mass of $N_2O$ added in the indirect transfer vessel additions to prepare the 500 µmol mol$^{-1}$ $N_2O$ intermediate and 325 nmol mol$^{-1}$ $N_2O$ reference materials, which combine to 73.07 % of the combined expanded uncertainty. There is a 3 mg uncertainty on the mass added for each direct gas addition ($N_2$, $O_2$, Ar), which provides a negligible contribution to the expanded uncertainty. The uncertainty in the $N_2O$ impurity in $O_2$ and $N_2$ provide contributions of 6.15 % and 17.87 %
respectively to the combined expanded uncertainty. The uncertainty contribution from the $N_2O$ impurity in the matrix gas scales with the amount fraction of the matrix component. The $N_2O$ impurity in Ar, and the uncertainty in relative molar masses provides negligible contributions to the combined expanded uncertainty.





### 4.1.2 Uncertainty in the amount fraction of reference materials for validation measurements

The amount fraction of $N_2O$ in a prepared mixture was validated through comparison to NPL in-house PRMs. The PRMs used for validation were derived from different parent mixtures, providing independence. In-house PRMs and unknown mixtures were measured alternately for ten-minute periods. To determine the certified amount fraction of the unknown mixture ($Y$), the gravimetric amount fraction of the in-house PRM ($X_1$) is multiplied by the mean ratio in analyser response ($X_2$), as shown in equation 5. Four repetitions provide four distinct measurements of this ratio.

$$Y = f(X_1 X_2) \tag{5}$$

Input quantities ($X_1$, $X_2$) have associated uncertainties that are combined to give a combined standard uncertainty for the measurement of $N_2O$ amount fraction derived from each validation. The standard uncertainty associated with the gravimetric amount fraction $u(x_1)$ is provided by the software Gravcalc2 (Brown, 2009). The standard uncertainty in the ratio measurement $u(x_2)$ is the standard deviation of the mean of the four ratios. Both input quantities were modelled with normal distributions and sensitivity coefficients ($c_1$, $c_2$) were taken as the partial derivatives with respect to each input quantity.

$$c_1 = \frac{\partial f}{\partial x_1} = x_2 \quad (6) \qquad c_2 = \frac{\partial f}{\partial x_2} = x_1 \quad (7)$$

Standard uncertainties were multiplied by respective sensitivity coefficients and combined in quadrature to provide a combined standard uncertainty for each validation. To combine the standard uncertainty from each validation, a sensitivity coefficient ($c = \frac{1}{3}$) was applied to each, providing equal weighting to the final analytical uncertainty.

Expanded analytical uncertainties of 0.07 % ($k$=2) were demonstrated using this approach. The final combined expanded uncertainty contains the contributions from gravimetric and analytical uncertainty. The combined expanded uncertainty of ambient amount fraction $N_2O$ reference materials is calculated to be 0.11 % ($k$=2) or 0.36 nmol mol$^{-1}$.

### 4.1.3 Reproducibility of reference gas production

The WMO-GAW has published an amount fraction range of 325-335 nmol mol$^{-1}$ representative of the unpolluted troposphere, while the range of $N_2O$ amount fractions covered by the WMO scale is somewhat broader (260-370 nmol mol$^{-1}$). The linearity of the CRDS analyser response to changes in amount fraction and the influence of amount fraction on apparent isotope delta values were investigated in the amount fraction range 320 to 360 nmol mol$^{-1}$ using a set of gravimetric prepared reference





materials. The lower boundary for amount fraction was defined by the CRDS analyser (Picarro G5131-*i*) $N_2O$ amount fraction

range for measurement of delta values.

The gravimetrically prepared reference materials were validated against a reference material prepared at nominally 325 nmol

mol$^{-1}$. Figure 3 shows the residual of the linear regression of the certified amount fraction as a function of the gravimetric

amount fraction for each reference material. The deviation from the linear regression does not show any obvious trend with

gravimetric amount fraction and falls within the extended WMO-GAW compatibility goal for all reference materials of ± 0.3

nmol mol$^{-1}$, demonstrating the suitability and linearity of the CRDS analytical technique for certifying $N_2O$ reference materials

in this range and the reproducibility of the reference materials produced.

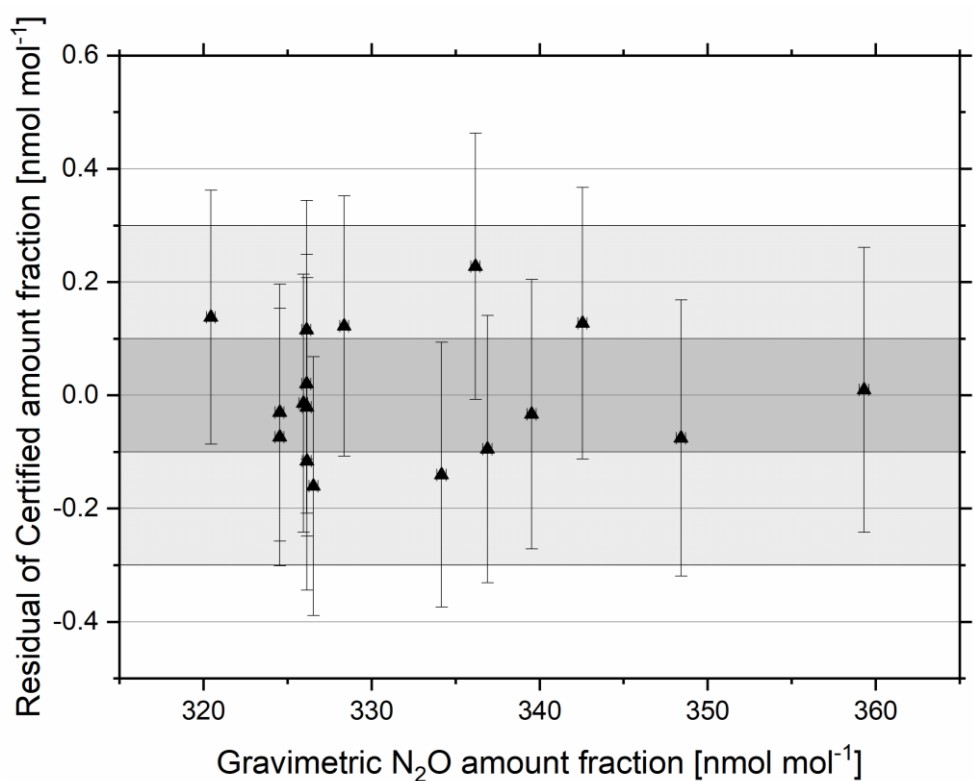


**Figure 3: Residuals of the linear regression of the certified amount fraction as a function of the gravimetric amount fraction for $N_2O$**
**reference materials in the amount fraction range 320-360 nmol mol$^{-1}$ certified against a nominally 325 nmol mol$^{-1}$ reference material.**
**The WMO-GAW Data Quality Objective (DQO) (± 0.1 nmol mol$^{-1}$) is indicated within the dark grey shading and the extended DQO**
**(± 0.3 nmol mol$^{-1}$) is indicated within the lighter grey shading. Error bars represent the combined (*k*=2) analytical (y-axis) and**
**gravimetric (x-axis, not visible) uncertainty.**



## 4.2 Stability of N$_2$O reference materials for amount fraction and isotopic composition

The demonstration of stability is important to achieve measurements of amount fraction and isotope ratio in the field with low uncertainty and also safeguards against drift in measurements as a result of changes in the reference material. The effect of
storage of reference materials of N$_2$O in synthetic air, with and without other greenhouse gas components in cylinders with different surface treatments was investigated.

### 4.2.1 Stability of reference materials for extended storage times

The stability of a nominally 325 nmol mol$^{-1}$ N$_2$O in synthetic air reference material was assessed over a three year period by
comparison with freshly prepared binary reference materials comprising N$_2$O in synthetic air prepared in the amount fraction range 300-360 nmol mol$^{-1}$ and reference materials containing N$_2$O in synthetic air and trace gases CO$_2$ (290-800 µmol mol$^{-1}$), CH$_4$ (1.8-3.0 µmol mol$^{-1}$) and CO (0.07-1.00 µmol mol$^{-1}$) (Figure 4). All validations within this period demonstrated agreement of amount fraction within the extended WMO compatibility goal of ± 0.3 nmol mol$^{-1}$ and thus demonstrates the stability of the 325 nmol mol$^{-1}$ N$_2$O reference material. The linearity of response of the CRDS in this amount fraction range is detailed above.
No clear distinction between agreement of validation of N$_2$O in synthetic air and N$_2$O within multi-component gas mixtures was observed, suggesting minimal interference of these gases on the CRDS analyser response to the total reported amount fraction of N$_2$O. The findings show agreement with those of (Erler et al., 2015; Harris et al., 2020) where no significant effect of CH$_4$, CO or CO$_2$ at atmospheric amount fraction was found on the reported N$_2$O amount fraction. In contrast, the authors reported a strong effect of O$_2$ amount fractions on apparent N$_2$O amount fraction, which they attribute to changes in the pressure
broadening. Similar effects on the CO$_2$ and CH$_4$ reported amount fractions with changing matrix composition when using CRDS have been observed earlier by (Nara et al., 2012).

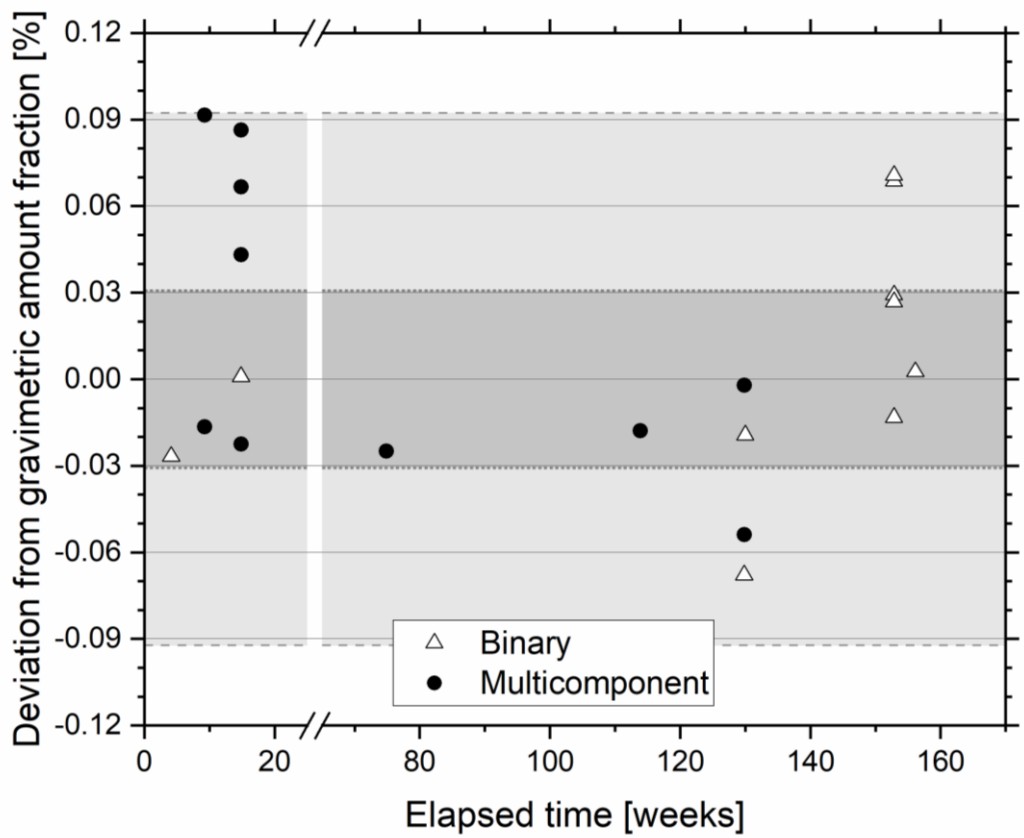

**Figure 4: Percentage difference between gravimetric and certified $N_2O$ amount fraction of a 325 nmol mol$^{-1}$ reference material as a**
**function of storage time. The reference material was certified against freshly prepared $N_2O$ reference materials in the amount fraction range 320-360 nmol mol$^{-1}$. The freshly prepared reference materials were either diluted in synthetic air (open triangle) or containing other greenhouse gases (filled circle). The WMO-GAW compatibility goal is indicated on the plot in dark grey shading and the extended compatibility goal is indicated by lighter grey shading.**



### 4.2.2 Stability of reference materials with reducing cylinder pressure

**Figure 5: Temporal change in (a) N₂O amount fraction and delta values: (b) δ¹⁸O, (c) δ¹⁵Nᵅ, (d) δ¹⁵Nᵝ in response to changes in cylinder pressure of four 0.85 L aluminium cylinders. Each data point represents a five-minute average of analyser response, error bars represent one standard deviation across the five-minute average.**

Figure 5 shows 5-minute averages for amount fraction, $\delta^{15}N^{\alpha}$, $\delta^{15}N^{\beta}$ and $\delta^{18}O$ with reducing pressure. No statistically relevant trend for N₂O amount fraction or delta value was observed as the cylinder pressure reduces. The experiment was conducted over 1.5 hours. A drift correction was conducted on the N₂O amount fraction and isotope delta values through subtraction of analyser response against a linear regression of the N₂O parent cylinder analysed immediately before and after each of the 0.85





L cylinders. There is good agreement within the standard deviation of the 5-minute responses between the amount fraction and delta values of the four cylinders and the parent cylinder. The data indicates that adsorption onto cylinder walls causes negligible changes in amount fraction and delta value with pressure. The findings show agreement with (Lushozi et al., 2019; Brewer et al., 2019) where no adsorption losses were identified in cylinder-to-cylinder transfer of a nominally 330 nmol mol[-1] reference material.

### 4.2.3 Cylinder treatments for enhanced stability with pressure

Figure 6 shows the 5-minute average of the response of $N_2O$ amount fraction with pressure relative to initial amount fraction for the three cylinders with different passivation processes. There is no difference in the reported $N_2O$ amount fraction between the three passivation processes. The data demonstrates that the internal passivation process causes negligible changes to the $N_2O$ analyser response for amount fraction with changes in cylinder pressure. The water vapour content of similar mixtures in synthetic air was measured to be around 0.5 µmol mol[-1] (Hill-Pearce et al., 2018).

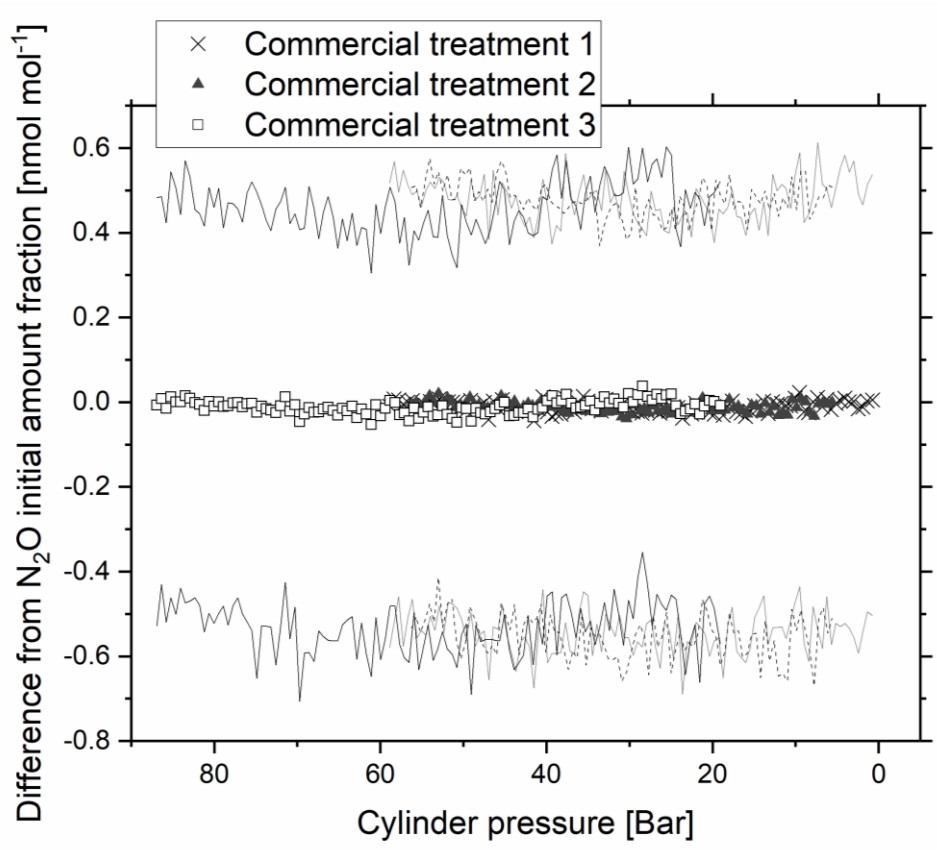



**Figure 6: Difference from initial N₂O amount fraction with reducing cylinder pressure for three nominally 325 nmol mol⁻¹ N₂O in synthetic air reference materials produced in 10 L cylinders with three different commercially available cylinder passivation processes. The uncertainty (one standard deviation) is plotted as dotted, dashed and solid lines.**

## 4.3 Absence of fractionation effects for static mixture production

The effects of the production and storage of ambient amount fraction N₂O in synthetic air reference materials in cylinders on the reported amount fraction and delta value were compared to reported vales for dynamic reference materials produced from the same pure N₂O source diluted with synthetic air (Figure 7). Dynamic reference materials demonstrate reduced adsorption effects compared to static standards particularly for low amount fraction reference materials (Platonov et al., 2018). Differences in reported delta value for static and dynamic standards of the same amount fraction would indicate fractionation events in
either production method.

The generated dynamic reference materials were validated against static reference materials of a similar amount fraction. The residuals of the static and dynamic linear regression of analyser response as a function of increasing gravimetric amount fraction of dynamic reference materials are shown in Figure 7. Agreement within 0.05 % (0.16 nmol mol⁻¹) was achieved
between the static and dynamic reference materials at nominally 325 nmol mol⁻¹.

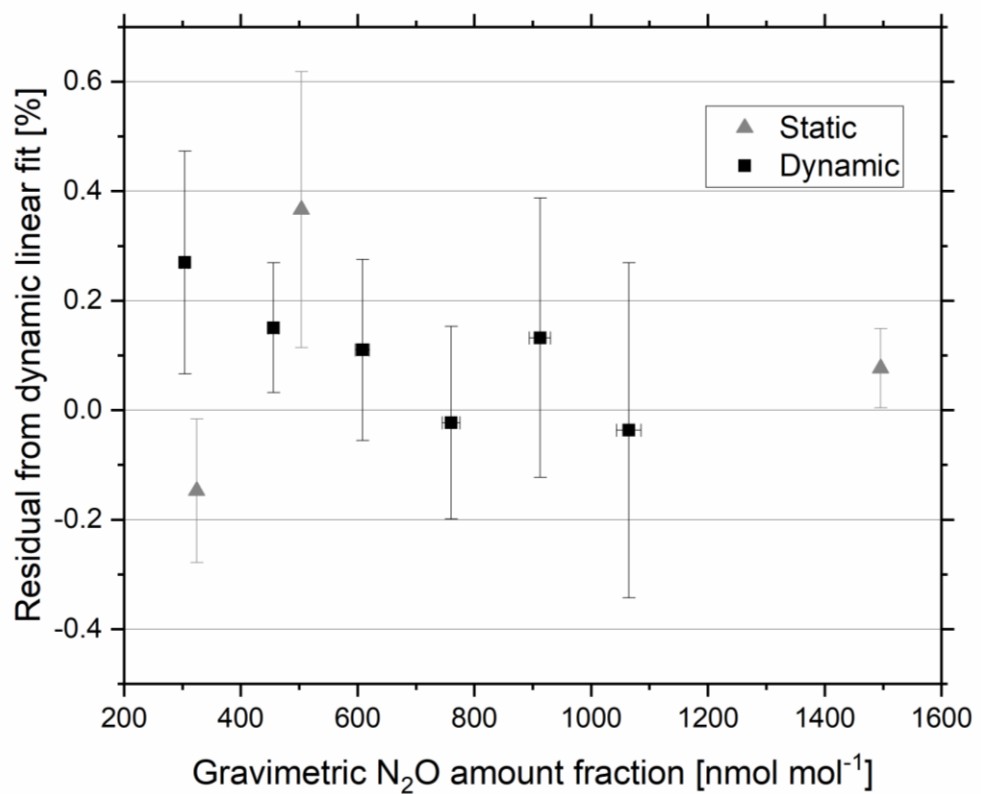

**Figure 7: Residuals from the linear regression of analyser response with increasing gravimetric amount fraction of the dynamic reference materials. Static (triangles) and dynamic (squares) reference materials in the amount fraction range 300-1500 nmol mol$^{-1}$ $N_2O$ in synthetic air. Error bars represent the gravimetric uncertainty ($k=2$) (x-axis) and the % reproducibility between the four repetitions of a five-minute average (y-axis).**

## 4.4 Comparison with existing scales

Two comparisons of amount fraction were carried out between NPL and the WMO/GAW World Calibration Centre (WCC-Empa) on reference materials prepared at Empa and NPL.

In a first approach, five reference materials were prepared by Empa containing $N_2O$ in the amount fraction range 290-370 nmol mol$^{-1}$ and certified against reference materials on the WMO-X2006A calibration scale (Hall et al., 2007; Noaa/Esrl, 2011) via quantum cascade laser absorption spectroscopy (QCLAS, model: QC-TILDAS-CS, 2200 cm$^{-1}$, Aerodyne Inc., USA). The Empa reference materials contained greenhouse and reactive gas components $CO_2$ (360-800 µmol mol$^{-1}$), $CH_4$ (1.7-3.2 µmol





mol$^{-1}$) and CO (120-560 nmol mol$^{-1}$). The reference materials were validated over a period of 6 months prior to analysis at NPL and were re-validated afterwards at Empa with a linear interpolation applied to account for any drift in amount fraction.

The reference materials were certified at NPL via CRDS against NPL in-house PRMs static reference materials in the amount

fraction range of 325-360 nmol mol$^{-1}$. Each sample was averaged across four repetitions measured of 10 minutes. Agreement within the WMO - GAW compatibility goal was achieved for amount fractions at nominally 330 nmol mol$^{-1}$ (Figure 8) and within the extended WMO - GAW compatibility goal at nominally 337 nmol mol$^{-1}$.

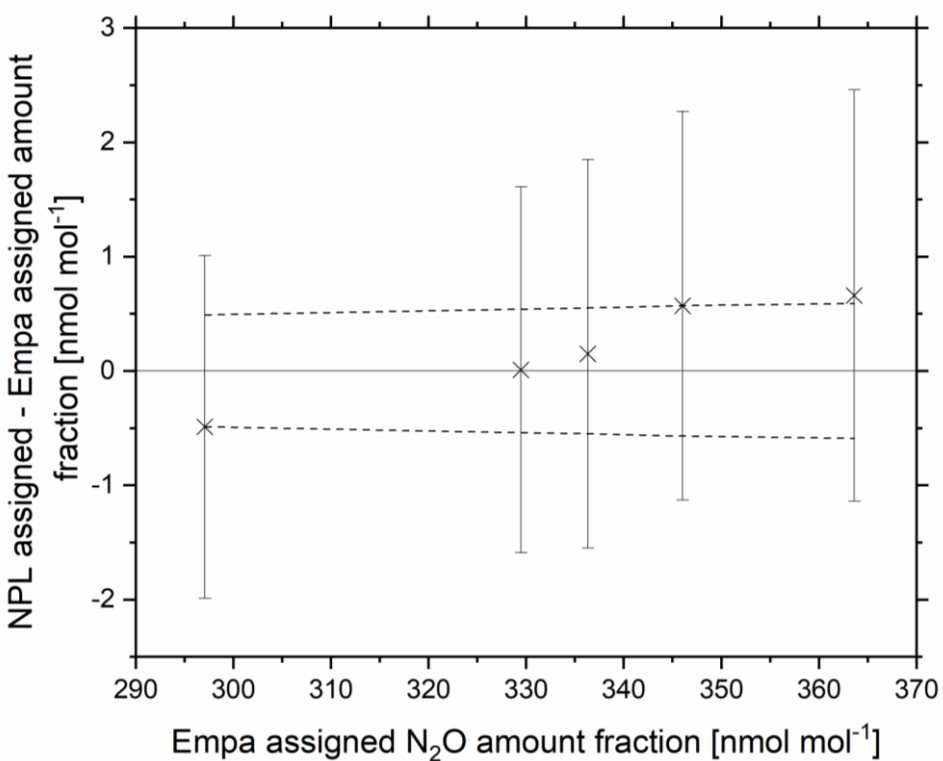

**Figure 8: Certified NPL amount fraction minus the certified Empa amount fraction of five N₂O reference materials in the range 290-370 nmol mol$^{-1}$ (crosses). Error bars represent the extended combined standard uncertainty ($k=2$) for NPL certified amount fraction. The Empa certified amount fraction is marked by a solid line at $y=0$ dashed lines represent the Empa measurement uncertainty.**

In a second approach, a nominally 325 nmol mol$^{-1}$ N₂O in synthetic air reference material containing nominally 526 nmol mol$^{-1}$ CO was prepared at NPL using the method described above and analysed at the World Calibration Centre (WCC-Empa) against NOAA/GMD reference materials on the WMO-X2006A calibration scale. Validations were performed via quantum cascade laser absorption spectroscopy (QCLAS, model: QC-TILDAS-CS, 2200 cm$^{-1}$, Aerodyne Inc., USA).





Agreement within the combined gravimetric uncertainty (*k*=2) was achieved for the nominally 325 nmol mol$^{-1}$ reference material (Figure 9). The uncertainty in the analytical amount fraction certified by Empa combines uncertainty contributions from traceability to the NOAA scale, scale propagation and repeatability the analytical system from analyser drift and pressure changes. For the Aerodyne analyser used in the comparison to validate the nominally 325 nmol mol$^{-1}$ N$_2$O reference material, the sources of uncertainty were combined to the combined standard uncertainty as shown in equation 8.


$$u_{N_2O} = [(0.06 \; nmol \; mol^{-1})^2 + (8.13e - 4 \times c)^2]^{1/2} \tag{8}$$

The expanded uncertainty (*k*=2) was determined from the standard uncertainty as shown in equation 9.


$$U_{N_2O} = 2 \times u_{N_2O} \tag{9}$$

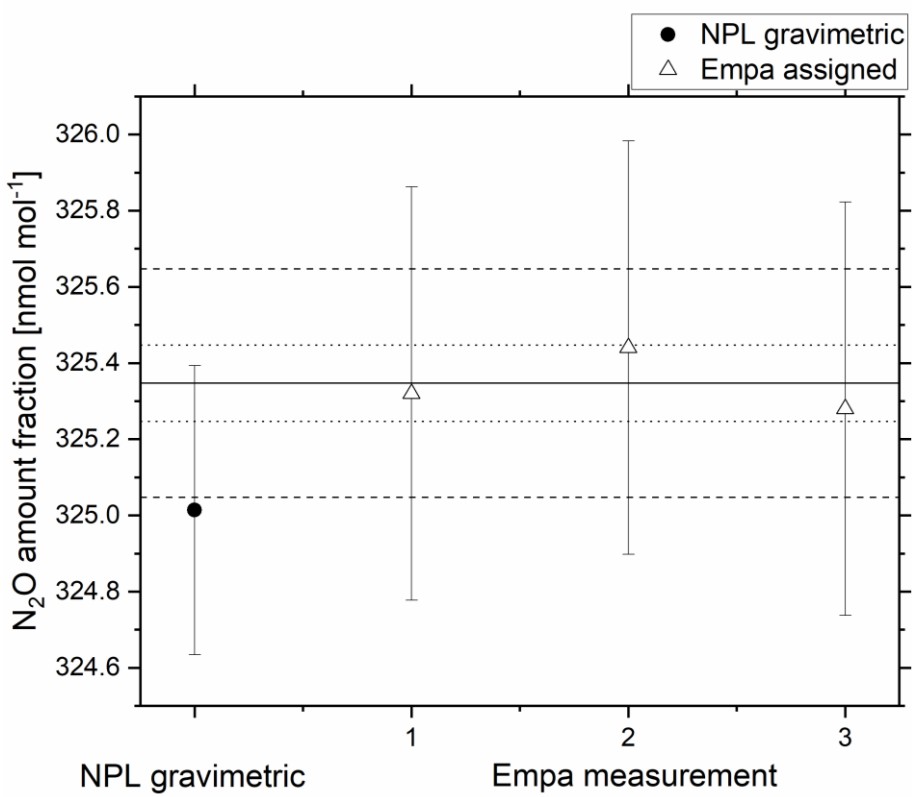

**Figure 9: Gravimetric amount fraction (filled circle) and amount fraction certified by Empa (open triangle) against the NOAA/GMD**

**reference scale for a nominally 325 nmol mol$^{-1}$ reference material. Solid black line represents averaged value certified by Empa, the**



dotted lines represent the WMO-GAW DQO and the dashed lines the extended WMO-GAW DQO. Error bars represent the combined expanded uncertainty (*k*=2).

## 5. Summary

N$_2$O reference materials with low uncertainty in amount fraction and isotope ratio are required for atmospheric monitoring. The stability of these reference materials is crucial to achieve these low uncertainties. We have demonstrated the production of atmospheric amount fraction N$_2$O reference materials with a gravimetric uncertainty within the WMO DQO for compatibility. Repeatability of the amount fraction of these reference materials was also within $\pm$ 0.3 nmol mol$^{-1}$. Gravimetry is the largest source of uncertainty for these reference materials, reducing the uncertainty further would require the use of lower

uncertainty balances for the indirect transfer of N$_2$O. The effect of including other greenhouse gases at atmospheric amount fractions in the reference materials did not significantly affect the amount fraction recorded by CRDS. Agreement between static and dynamic reference materials of 0.05 % was achieved between reference materials at nominally 325 nmol mol$^{-1}$.

The amount fraction of a prepared N$_2$O reference material in synthetic air with atmospheric amount fraction of CO was

compared to internationally recognised scales (WMO-X2006A) and found to be in agreement within the gravimetric uncertainty. The amount fraction of reference materials produced at Empa were measured at NPL through comparison with NPL in-house PRMs. For a nominally 330 nmol mol$^{-1}$, the NPL and Empa assigned amount fraction values were in agreement within the WMO DQO for compatibility of $\pm$ 0.1 nmol mol$^{-1}$ and the full range of PRMs (nominally 290-370 nmol mol$^{-1}$) were in agreement within the measurement uncertainty of nominally $\pm$ 0.5 nmol mol$^{-1}$.


The change in amount fraction of the mixtures with reducing cylinder pressure was shown to be smaller than measurement uncertainty regardless of cylinder passivation chemistry and the stability of the mixtures over three years was within the expanded WMO-GAW DQO for compatibility. The isotopic composition of the reference mixtures was also demonstrated to be stable with reducing pressure, and agreement of delta values was achieved for static reference materials with dynamic

dilutions within the analytical uncertainty. The next steps towards producing reference materials for source apportionment will be to produce reference materials with a range of isotopic values and to verify the assignment of their delta values.

ACKNOWLEDGEMENT






The authors would like to acknowledge the financial support for the 16ENV06 SIRS project that was provided by European Metrology Programme for Innovation and Research (EMPIR) programme co-financed by the Participating States and from the European Union's Horizon 2020 research and innovation programme.

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
