# Peer review of "Characterisation of Gas Reference Materials for Underpinning Atmospheric Measurements of Stable Isotopes of Nitrous Oxide"

_Atmospheric Measurement Techniques, 2021_

## Author Response (AR2)

- **RC1**: 'Comment on amt-2021-45', Stefan Persijn, 17 Mar 2021 reply

The authors describe N2O gas standards prepared by gravimetric methods at ambient levels and with low uncertainty and they provide data on the isotopic ratios and the long-term stability. Such N2O gas standards are of key importance for atmospheric research and in particular the provided data on the isotopic ratios makes the paper a good contribution to existing literature.  Therefore, I suggest that the paper is accepted. Some minor points:

1. The introduction section can be improved. There are quite some equations and definitions and their need is not always clear (e.g. site-specific distribution expressed by SP in equation 4).

   This has been addressed throughout the introduction text.

   The equation for site preference and bulk $\delta^{15}N$ has been removed from the introduction.

2. A table with an uncertainty budget for the developed standards can help the reader get a quick overview of the main uncertainty contributions. From the current text this is more difficult to grasp.

   Please see the added table at line 247 which contains the sources of gravimetric uncertainty which replaces the description in the text.

3. line 115 'to inhibit adsorption of target components' Is there any evidence in literature that N2O at these amount fractions adsorbs on aluminium? Regarding the other components in the mixtures (except for CO2 for which adsorption has been demonstrated) is there some evidence that they adsorb?

   To the best of the authors' knowledge there is no published evidence that $N_2O$ adsorbs on aluminium or passivated surfaces. We expect to provide $N_2O$ reference materials in combination with other greenhouse gases and, as such, appropriate passivation chemistry to reduce adsorption of e.g. $CO_2$ should be selected and tested for use with $N_2O$.

4. 'Figure 5: For some symbols only a cap but no error bar is shown.

   The formatting of the error bars has been updated in figure 5 so that they are clearer to see.

5. Line 125: '± 0.3 mg' and Line 129: '± 3 mg' Please give some explanation.

   Line 116: '± 0.3 mg' and Line 121: '± 3 mg'. This describes the mass uncertainty on the two balances we use during the preparation of our gaseous reference materials. 0.3 mg is the mass uncertainty on the balance we use to weigh the transfer vessels

used for indirect transfer vessel additions. The 3 mg is the mass uncertainty for the balance we use to weigh 10 litre gas cylinders between direct gas additions.

6. line 262 'providing independence' Rephrase as within one lab often parent mixtures do have some dependencies (e.g. prepared using the same instruments, or purity analysis of the pure gases using the same analytical instruments and reference gases)

We have rephrased as follows;

line 252: "The amount fraction of $N_2O$ in a prepared mixture was validated through comparison to NPL in-house PRMs. The PRMs used for validation were derived from different parent mixtures *which are, where possible, produced by different operators in order to provide a greater degree of independence from errors in amount fraction of a parent mixture.*"

7. Line 27 '(WMO, 2020)' seems to be missing from the list of references.

Updated in the references, it was previously listed with just the website address.

8. A few typos, e.g. line 366 'vales', line 395 'measured of 10 minutes', figure 3 'of Certified amount'

The spelling errors have been corrected in the text. See lines 366 (values), line 392 (10 minute repetitions) and figure 3 axis (certified amount).

9. In equation 8, where does 'c' refers to?

In equation 8 (now 6 with updates to introduction), 'c' refers to the amount fraction of N2O, in nmol mol-1. The term '8.13e-4' refers to the combined uncertainty from the NOAA scale and pressure changes. The term '0.06 nmol mol-1' refers to the combined uncertainty from scale propagation and analyser drift. A description of the terms has been added to the text. See line 421.

10. In figure 8, there seems to be a clear trend in the assigned values of NPL - Empa as a function of the Empa assigned amount fraction. Please comment on this.

The following text was added;

Line 397: Reference materials with amount fractions within the range 295- 345 nmol mol$^-$$^1$ verified within the experimental extended combined standard uncertainty ($k$=2) for NPL certified amount fraction. A trend was observed with lower NPL amount fractions certifying lower and higher NPL amount fractions certifying higher.

11. Lines 250-254: Mismatch between the number of significant digits for the mass uncertainty ('0.3 mg', '3 mg') and the contributions to the percentage of the combined uncertainty ('73.07 %' '17.87 %').

The number of significant figures for the percentage contributions in table 1 has been reduced to align with the significant figures in the mass uncertainties.

12. Correct the author list of reference 'BIPM et al., 2008'.

The author list has been updated to include all of the institutions.

13. Line 148-150. It is mentioned that N2O was determined in the matrix gas but no data are presented. Please provide these data including the corresponding uncertainty.

The $N_2O$ in the synthetic air matrix was determined as $0.75 \pm 0.09$ nmol mol$^{-1}$. This value, corresponding uncertainty and a reference to the method of determination have been added to the text, see line 142. There is ongoing research to improve the accuracy of the quantification which may form the basis of a future publication.

**Citation**: https://doi.org/10.5194/amt-2021-45-RC1

14. **RC2**: 'Comment on amt-2021-45', Anonymous Referee #2, 19 Mar 2021 reply

This paper describes the characterization of compressed gas reference materials containing nitrous oxide, and their application to measurement of atmospheric N2O and its stable isotopes. Of particular importance is the finding that dynamic dilution does not appear to be impacted by isotopic fractionation. This work confirms previous work and offers new insights into stability as a function of cylinder pressure. This work supports on-going efforts to better understand and implement measurements of N2O and stable isotopes of N2O.

General Comments

While you do describe the gravimetric process and compare N2O to the WM/GAW N2O scale, you do not offer an assessment of new isotopic reference materials (USGS51 and USGS52) or alternate sources of stable isotopes of nitrogen and oxygen and application to N2O. Therefore, I suggest the following title better described this work.

Characterization of Gas Reference Materials for Underpinning Atmospheric Measurements of Stable Isotopes of Nitrous Oxide

The title has been changed to reflect your suggestion.

Some readers may be unfamiliar with the terms "validation" and "certification". It might help to expand on what these terms. For example, for certification, equation (5) indicates that a single primary standard is used to "certify" (or value-assign) a mixture. Is that primary standard part of a suite of standards that define a "scale", or is each PRM unique in its application to "certify" mixtures? Is the validation process simply comparing a new gravimetric standard to others to verify consistency, or is there more to it? Can a gravimetric standard have both a gravimetric value and a "certified value"? For example,

in figure 3 you state that this shows the residuals of certified amount fractions vs gravimetric amount fractions.  Is this the same as the residuals from a linear fit of analyzer response vs gravimetric amount fraction, which provides a measure of consistency of gravimetric preparation?  I guess I don't understand why one would "certify" mixtures against a single 325 ppb reference material when you appear to have a consistent set of gravimetric standards that could be used to define a "scale".  If that is in fact what you have done, then perhaps this could be explained more clearly.

Each standard has a gravimetrically assigned amount fraction, providing traceability to the SI and offering an approach for widespread comparability.

Each PRM is unique (not part of a scale), therefore the **certified amount fraction** changes depending on the standard used for reference. Figure 3 shows the residuals (difference between gravimetric and certified) for several standards all with reference to a single standard. This demonstrates the consistency in the gravimetric assignment over all standards prepared and across an amount fraction range.  The reference materials could be used to define a scale, however we aim to highlight the advances in the preparation of the PRM itself.

What are the terms in in equation 8 and where do they come from?

In equation 8 (now equation 6 with the updated introduction), 'c' refers to the amount fraction of $N_2O$, in nmol mol$^{-1}$. The term '8.13e-4' refers to the combined uncertainty from the NOAA scale and pressure changes. The term '0.06 nmol mol$^{-1}$' refers to the combined uncertainty from scale propagation and analyser drift. A description of the terms has been added to the text.

Figure 5:  I'm not clear what the "parent" has to do with this.  Why not show X(P) - X(Po) where X(P0) is the amount fraction at the initial pressure (as in Fig 6)?  Importantly, it looks like one of the cylinders in "a" is changing (triangles). It shows a decrease of ~0.2% from start to finish, certainly the CRDS can detect that level of change.

The authors feel that the amount fraction and delta value of the parent is of interest as it shows that the amount fraction and delta values do not change between the parent and the daughter cylinders on decant. The value of the parent cylinder is demonstrated to be very close to the values at the initial pressures. While the CRDS can detect small changes in amount fraction, it was challenging to record enough data so that trends can be observed with as little analyser noise as possible. In figure 5 the authors found an optimum averaging time of 5 minutes of data recorded at ~0.3Hz, which gave more averaged data points than longer averaging times making trends easier to observe. The changes in values observed when venting the cylinders are much smaller than the standard deviation of the 5-minute averages.

Figure 6:  It seems like the uncertainty plotted is not relevant here.
We have now removed the uncertainty from the plot.

Clearly the repeatability is very good, and so we can conclude that the amount fractions are stable to within +/- 0.05 ppm (maybe less) in these cylinders.

The authors agree with this conclusion. The following has been added to line 355;

"There is no difference in the reported $N_2O$ amount fraction between the three passivation processes, *with all stable to within 0.05 nmol mol$^{-1}$.*"

Also, why are these data so much more precise than N2O amount fractions in Fig 5a? Here the scatter for all points is < 0.05 ppb, whereas in fig 5a it is ~0.1%, or about 0.3 ppb.

The RSD across the 5 minute averages for amount fraction is nominally the same for the data in figure 5a and figure 6.. In figure 6, the valuesare scaled to each cylinder's initial amount fraction, so the axis values are closer to zero than figure 5a, where the four cylinders are scaled to the parent amount fraction. Each of the smaller cylinders has a different initial deviation from the parent cylinder, which makes the spread of data look larger. Taking the smaller cylinders individually, the precision is on the same scale as for figure 6.

Figure 7: Is the linear regression fit based on just the dynamic standards or both dynamic and static standards? It is odd that the mean residual does not appear to be "zero".

The linear regression was calculated from the dynamic standard gravimetric concentration against analyser response. The static and dynamic analyser response was subtracted from this dynamic linear regression. We have re-processed the data using calculations which only use the plotted data. Please see figure 7. The mean residual of the dynamic data points is now "zero".

Figure 8: Do the NPL PRMs represent a defined NPL scale that can expected to be maintained in a consistent manner over time?

In the work presented here, NPL has not aimed to produce an NPL $N_2O$ scale but rather realise the primary standard and the associated uncertainty which can be compared under CIPM MRA. This approach offers benefits over a scale approach such as long term stability and the ability to reproduce the standard within its stated uncertainty at any time. The authors understand that there are also benefits to the measurement community of a scale approach such as improved precision and measurement compatibility between comparisons to the scale artefact. This may be preferable where the key quantity to be measured is the relative difference in amount-of-substance fraction within a measurement network.

Specific Comments

78: I suggest:  "… this technique has demonstrated a precision (Allan deviation) of < 0.05 nmol/mol …"

Comment addressed in the text, please see line 69.

96:  replace ("Scott Marrin") with "(Scott Marrin, now Praxair)"

Comment addressed in the text, please see line 87.

114:  Is "electropolishing" correct?  I am not familiar with electropolished Luxfer cylinders.  If electropolishing is correct, please specify the company that performed the electropolishing.

The cylinders have Luxfer's SGS™ (Superior Gas stability) internal surface. The name of the cylinders has been updated in the text.

Information on the Luxfer website: "The SGS interior is achieved by processing a cylinder through a series of proprietary, time-sensitive manufacturing operations that produce a consistent, better-performing internal surface."

122:  How was gas from the transfer vessel introduced?  By flushing or expansion to vacuum?

Gas is transferred from the transfer vessel to the cylinder through expansion to vacuum. The gas cylinder is evacuated overnight to pressures < 5 x10-7 mbar before transfer vessel addition. All the connections between the cylinder and the transfer vessel are also evacuated, to pressures of < 1x10-5 mbar.

347:  Venting a cylinder in 1.5 hours would also result in significant cooling and potential thermal fractionation, so it seems that your tests suggest that adsorption AND thermal fractionation of N2O are less than the level of detection, except for the one cylinder in fig. 5 that appears to show a change in N2O.

The cylinders which were vented in 1.5 hours are 0.85 litre water volume cylinders containing less than 30 litres of gas when filled. The venting of the cylinders was carried out within the flow rate range required to achieve optimum pressure through the analyser. As such, we do not expect to observe fractionation.

The following text is added to ensure this is clear to the reader;

Line 338: "During venting of the cylinder through the analyser a small excess flow rate of between 0.3 and 0.5 litres per minute was maintained to optimise the pressure through

the analyser, reduce back diffusion into the analyser and ensure thermodynamic effects do not occur."

392: You say you used linear interpolation to account for drift.  Is this at odds with experiments that suggest stability over time and with changing pressure?

The line at 356 reads "A drift correction was conducted on the $N_2O$ amount fraction and isotope delta values through subtraction of analyser response against a linear regression of the $N_2O$ parent cylinder analysed immediately before and after each of the 0.85 L cylinders"

Analyser drift requires monitoring and accounting for with regular comparison to the value of a reference material. All measurements were regularly compared to in house primary reference materials.

395:  delete "measured"

Comment addressed in the text, please see line 395.

425: replace "averaged" with "average"

Comment addressed in the text, please see line 431.

Figure 5:  Figures are not labelled "a", "b", etc.

The figures in figure 5 have been labelled as "a", "b", etc. Please see line 330.

Figure 6:  Were these tests also performed over a short period, similar to those in Fig. 5?

The measurements shown in figure 6 took longer to complete as the volume of the cylinders in larger (10 litres) than the data shown in figure 5 (0.8 litres) and the cylinders were filled to higher pressure in figure 6. The excess flow rate not passed through the analyser was maintained at 0.5 litres per minute in order to achieve optimum pressure for the analyser and so as not to cause any thermodynamic effects observed at higher flow rates.

Missing reference to "WMO (2020)"

Updated in the references, it was previously listed with just the website address.

540:  Is this a reference to a WMO Greenhouse Gas Bulletin?,  e.g. https://Public.wmo.int/en/resources/library/wmo-greenhouse-gas-bulletin

This is a reference for the WMO GHG Bulletin No. 15. I have added the full name and the link to the reference.

541:  The link appears to be out of da

The links have been checked and are up-to-date